# Co-Inoculation with *Azospirillum brasilense* and *Bradyrhizobium* sp. Enhances Nitrogen Uptake and Yield in Field-Grown Cowpea and Did Not Change N-Fertilizer Recovery

**DOI:** 10.3390/plants11141847

**Published:** 2022-07-14

**Authors:** Fernando Shintate Galindo, Paulo Humberto Pagliari, Edson Cabral da Silva, Vinicius Martins Silva, Guilherme Carlos Fernandes, Willian Lima Rodrigues, Elaine Garcia Oliveira Céu, Bruno Horschut de Lima, Arshad Jalal, Takashi Muraoka, Salatiér Buzetti, José Lavres, Marcelo Carvalho Minhoto Teixeira Filho

**Affiliations:** 1Center for Nuclear Energy in Agriculture, University of São Paulo, Piracicaba 13418-900, Brazil; muraoka@cena.usp.br (T.M.); jlavres@usp.br (J.L.); 2Southwest Research and Outreach Center, Department of Soil, Water, and Climate, University of Minnesota, Lamberton, MN 56152, USA; pagli005@umn.edu; 3Goiano Federal Institute, Rio Verde Campus, Rio Verde 75901-970, Brazil; edsoncabralsilva@gmail.com; 4Department of Biology Applied to Agriculture, São Paulo State University, Jaboticabal 14884-900, Brazil; vmsagr@gmail.com; 5Department of Plant Health, Rural Engineering, and Soils, São Paulo State University, Ilha Solteira 1585-000, Brazil; guilherme.carlos.fernandes@gmail.com (G.C.F.); willianrodrigues53@gmail.com (W.L.R.); elaine.garcia@unesp.br (E.G.O.C.); bruno.horschut@unesp.br (B.H.d.L.); arshad.jalal@unesp.br (A.J.); sbuzetti@agr.feis.unesp.br (S.B.); mcmtf@yahoo.com.br (M.C.M.T.F.)

**Keywords:** plant growth rhizobacteria, biological nitrogen fixation, nitrogen-^15^, nitrogen fractions, *Vigna unguiculata* (L.) Walp

## Abstract

This study was designed to investigate the effects of *Azospirillum brasilense* and *Bradyrhizobium* sp. co-inoculation coupled with N application on soil N levels and N in plants (total N, nitrate N-NO_3_^−^ and ammonium N-NH_4_^+^), photosynthetic pigments, cowpea plant biomass and grain yield. An isotopic technique was employed to evaluate ^15^N fertilizer recovery and derivation. Field trials involved two inoculations—(i) single *Bradyrhizobium* sp. and (ii) *Bradyrhizobium* sp. + *A. brasilense* co-inoculation—and four N fertilizer rates (0, 20, 40 and 80 kg ha^−1^). The co-inoculation of *Bradyrhizobium* sp. + *A. brasilense* increased cowpea N uptake (an increase from 10 to 14%) and grain yield (an average increase of 8%) compared to the standard inoculation with *Bradyrhizobium* sp. specifically derived from soil and other sources without affecting ^15^N fertilizer recovery. There is no need for the supplementation of N via mineral fertilizers when *A. brasilense* co-inoculation is performed in a cowpea crop. However, even in the case of an NPK basal fertilization, applied N rates should remain below 20 kg N ha^−1^ when co-inoculation with *Bradyrhizobium* sp. and *A. brasilense* is performed.

## 1. Introduction

Cowpea (*Vigna unguiculata* (L.) Walp.) is an herbaceous legume crop showing considerable adaptation to tropical and subtropical climates and is cultivated across Africa, Latin America, Southern United States and Southeast Asia [1]. Cowpea is also cultivated in some Mediterranean countries, but cultivation is not widespread in Europe [2]. This legume crop is particularly important due to its high protein content, tolerance to drought and high temperatures, as well as tolerance to low fertility soils [3]. Being drought-tolerant, cowpea is grown predominantly in dry Savannahs around the world, with an annual rainfall of around 300 mm or even less [4]. Cowpea is cultivated mainly for grain production and is rich in protein with the most improved varieties containing, on average, 23% of protein on a dry weight basis [4]. Although beans are the primary focus of the cowpea plant, both leaves and flowers are also considered edible [1,5]. These aspects make cowpea a key crop in the context of global climate change and food security.

Cowpea belongs to the Fabaceae (or Leguminosae) family, which speaks for their capacity to fix atmospheric nitrogen (N_2_) [6]. Cowpea plants can form relationships with N-fixing bacteria in a process termed symbiotic biological N_2_ fixation (BNF). In this process, bacteria inhabiting plant roots form nodules within the roots, and those nodules can fix atmospheric N (by converting N_2_ to the usable NH_3_^+^ form via nitrogenase enzyme complex), which is utilized by the plant in exchange for carbohydrates [7,8]. However, the efficiency of the N-fixing activity of the symbiotic association between plants and bacteria varies among legume species and legume-symbiont combinations, and cowpea is often characterized as a poor N-fixer compared to other legume crops [9,10,11,12,13]. It has been estimated that the amount of N fixed in different bean species, including cowpea, was low, varying between 25 and 70 kg N_2_ fixed ha^−1^ [6,11,12,14], which represents a contribution between 50 and 80% of its N requirement [15,16]. These values are considerably lower than the amount of N fixed in soybean (*Glycine max* (L.) Merrill), which can range from 80 to above 300 kg N fixed ha^−1^ [17,18,19,20,21,22]. Cowpea may demand up to 100 kg N ha^−1^ per season [23], thus, N fertilizer application may be required to fulfill the crop demand depending on environmental conditions and yield expectations. The process of fertilizer addition is known for leading to a downregulation of BNF. Plants cannot take up all the N fertilizer applied leading to impaired N use efficiency and recovery and rising environmental concerns. Furthermore, BNF is a process vulnerable to drought, a common environmental condition under tropical agriculture, and drops even before the transpiration and photosynthetic rates decrease significantly [12].

In this sense, the co-inoculation with plant growth-promoting rhizobacteria (PGPR) can provide cowpea with several growth-promoting attributes, and thereby improve plant growth, enhancing N use efficiency and potentially increasing BNF [24,25,26,27,28,29,30]. Co-inoculation could be defined as the assemblage of at least two microorganisms that contribute to diverse microbial processes, improving plant growth and development [22]. A recent meta-analysis study with 42 published articles concluded that the co-inoculation technology with the most applied PGPR (*Azospirillum, Bacillus*, *Pseudomonas* and *Serratia* spp.) resulted in a significant increase in shoots (7%), roots (13%) and nodule biomass (6%) [31]. Co-inoculation with *Bacillus* sp. and *Bradyrhizobium* sp. was reported to enhance nodulation and N_2_-fixing efficiency by producing larger nodules [32] which led to better plant yield [33]. Although several PGPRs have been assessed for their potential to work in co-inoculation with *Bradyrhizobium* sp. in different legume crops, *Azospirillum brasilense* has stood out [22,34,35,36].

The free-living rhizobacteria (genus *Azospirillum*) are able to colonize more than one hundred plant species in almost every living soil on earth, providing conditions that improve crop growth, development and production [26,37,38]. The *A. brasilense* Ab-V5 and Ab-V6 strains are officially authorized by the Brazilian government for inoculant production with the aim to increase wheat and maize productivity under field conditions [24,38]. It has been reported that *A. brasilense* strains (Ab-V5 and Ab-V6) are carrying similar fix and nif genes that are contributing to BNF [25]. Despite this, these strains have a different capacity for phytohormone production [26] but still share similar genes for auxin synthesis. To date, the additive hypothesis best addresses the operating principle of *A. brasilense*, in which multiple mechanisms of plant growth promotion work in convergence or in sequence [30,39,40]. For example, *A. brasilense* has been reported to promote plant growth by increasing the production of phytohormones such as gibberellins, auxins and cytokinins [26]; increasing root development leading to greater nutrients and water acquisition [41]; enhancing BNF [42]; increasing N use efficiency from applied N fertilizer [28,29]; solubilization of phosphates [43]; among others.

We hypothesized that cowpea co-inoculation with *Bradyrhizobium* sp. and *A. brasilense* can increase N accumulation, leading to a reduction in the amount of N-based fertilizer needed for optimal productivity. Therefore, we aimed to access the potential benefits and action mechanisms of *Bradyrhizobium* sp. + *A. brasilense* co-inoculation coupled with N fertilizer rates in cowpea. The study focuses on the potential for increased N uptake and plant growth-related parameters under field conditions due to co-inoculation. Our main goal was to determine if this practice will reduce the amount of N-based fertilizers required for optimum cowpea productivity. To achieve this objective, the stable isotopes technique was employed to evaluate the ^15^N fertilizer recovery and derivation, in addition to N fraction in soil and plants targeting to reduce the use of N fertilizers in cowpea production systems.

## 2. Results

### 2.1. Soil and Root Nitrogen Fractions

Inorganic N (N-NH_4_^+^ and N-NO_3_^−^) and total N contents in soil and N-NO_3_^−^ in root were not greatly influenced by *A. brasilense* co-inoculation and N application rates, except soil N-NO_3_^−^, which was greater with high N input (80 kg N ha^−1^) relative to the control (without N application) in both *A. brasilense* co-inoculated and single *Bradyrhizobium* sp. inoculated plots (Figure 1A–C,F).

In the absence of N application and under low N input (20 kg N ha^−1^), *A. brasilense* co-inoculation provided greater root ureide and protein compared with single *Bradyrhizobium* sp. inoculation (Figure 1D,G). For both co-inoculated and single *Bradyrhizobium* sp. inoculation treatments, the increase in N application rates tended to reduce root ureide and soluble protein concentrations (Figure 1D,G). Regarding root N-NH_4_^+^ concentration, *A. brasilense* co-inoculation provided greater N-NH_4_^+^ compared with single *Bradyrhizobium* sp. inoculation regardless of N rate applied (Figure 1E). Root N-NH_4_^+^ showed the highest values with medium N input (40 kg N ha^−1^) when co-inoculation was performed, while root N-NH_4_^+^ was greater with medium N input (40 kg N ha^−1^) compared to high N input with single *Bradyrhizobium* sp. inoculation (Figure 1E).

### 2.2. Leaf Nitrogen Fractions and Chlorophyll Pigment

Leaf ureide, N-NH_4_^+^, total N and chlorophyll A + B concentrations were influenced by *A. brasilense* co-inoculation (Figure 2A,B,D,F). Without N application and under low N inputs, *A. brasilense* co-inoculation provided augmented leaf N-NH_4_^+^ and total N compared with single *Bradyrhizobium* sp. inoculation (Figure 2B,D). Regarding leaf ureide concentration, the control, low and medium N inputs, *A. brasilense* co-inoculation had greater ureide concentration compared with single *Bradyrhizobium* sp. inoculation (Figure 2A). Chlorophyll A + B was greater with *A. brasilense* co-inoculation regardless of the N rate applied (Figure 2F). Leaf N-NO_3_^−^ and soluble protein concentrations were not greatly affected by co-inoculation (Figure 2C,E).

The increase in N fertilizer rates decreased leaf ureide, N-NH_4_^+^, soluble protein and chlorophyll A + B concentrations (Figure 2A,B,E,F). Soluble protein decreased 33.6% under single *Bradyrhizobium* sp. inoculation (from 438 to 291 mg g^−1^) and 38.1% with *A. brasilense* co-inoculation (from 469 to 291 mg g^−1^) (Figure 2E). In contrast, leaf N-NO_3_^−^ was greater with high N input compared with control, low and medium N rates (Figure 2C), while leaf total N was not influenced by N rates (Figure 2D).

### 2.3. Biomass Production and Grain Yield

Roots, shoot biomass and grain yield responded positively to *A. brasilense* co-inoculation, especially in the absence of N or under low N inputs (Figure 3A,C,D). We verified an increase of 15, 13 and 21% (root, shoot biomass and grain yield, respectively) in co-inoculated plots compared with single *Bradyrhizobium* sp. inoculated plots in the absence of N application (Root: −Azo = 171 vs. +Azo = 196 kg ha^−1^; Shoot: −Azo = 4700 vs. +Azo = 5290 kg ha^−1^; Yield: −Azo = 1707 vs. +Azo = 2072 kg ha^−1^) (Figure 3A,C,D). With a low N input, we verified an increase of 17, 14 and 17% (root, shoot biomass and grain yield, respectively) in co-inoculated plots compared to single *Bradyrhizobium* sp. inoculated plots (Root: −Azo = 154 vs. +Azo = 181 kg ha^−1^; Shoot: −Azo = 4297 vs. +Azo = 4902 kg ha^−1^; Yield: −Azo = 1647 vs. +Azo = 1925 kg ha^−1^) (Figure 3A,C,D).

Nodule biomass was not greatly influenced by *A. brasilense* co-inoculation (Figure 3B). In general, all the evaluated productive components (root, nodule, shoot biomass and grain yield) tended to reduce with increasing N application rates (Figure 3).

### 2.4. Shoot and Root Nitrogen Accumulation and ^15^N-Fertilizer Recovery

Shoot N accumulation and shoot N derived from soil and other sources (SNDFS) were greater when *A. brasilense* co-inoculation was performed compared to single *Bradyrhizobium* sp. inoculation in the absence of N application (an increase of 11%) and with low N input (an increase of 16%) (Figure 4A,B). In addition, shoot N accumulation was 10% greater with *A. brasilense* co-inoculation coupled with medium N input relative to single *Bradyrhizobium* sp. inoculation (Figure 4A).

Similarly, grain N accumulation and grain N derived from soil and other sources (GNDFS) were greater when *A. brasilense* co-inoculation was performed relative to single *Bradyrhizobium* sp. inoculation regardless of N rate applied (Figure 4E,F). We verified an average increase of 13 and 14% (grain N and GNDFS, respectively) in co-inoculated plots compared to single *Bradyrhizobium* sp. inoculated plots (Figure 4E,F). Nitrogen derived from fertilizer (SNDFF and GNDFF) and ^15^N fertilizer recovery in both shoot and grain tissue were not greatly influenced by *A. brasilense* co-inoculation (Figure 4C,D,G,H).

Nitrogen accumulated in shoot and grain, SNDFS, GNDFS and ^15^N fertilizer recovery in both shoot and grain tissue tended to reduce with increasing N fertilizer rates (Figure 4A,B,D–F,H). In contrast, SNDFF and GNDFF were greater under high N input compared with low and medium N inputs (Figure 4C,G).

## 3. Discussion

Based on our results, *A. brasilense* co-inoculation with *Bradyrhizobium* sp. could be considered a best management practice that is effective at increasing cowpea N uptake, N metabolism, and grain yield compared to the single inoculation with *Bradyrhizobium* sp. We observed increases in cowpea N uptake, N metabolism, and grain yield in co-inoculated plants. Furthermore, the increase in root N content (mainly as ureide and N-NH_4_^+^) and root biomass accumulation, observed in co-inoculated plots, could be the key mechanism for the increase in root and leaf soluble proteins and the improved cowpea growth and yield.

The assessment of the root system has been the focus of several co-inoculation studies with *A. brasilense* in different legume crops. Most of these studies have reported increased growth of root hairs and lateral roots [22,44,45]. Interestingly, although plant nodule biomass was not affected by co-inoculation, we verified a numeric increase of 17% in nodule biomass when *A. brasilense* co-inoculation was performed (476 vs. 557 g ha^−1^ in single inoculated and co-inoculated plots, respectively). It is worth mentioning that the lack of significance may be due to the high variability of this variable; however, there is some evidence of a numerical increase in nodulation (nodule biomass).

The increased ureide and N-NH_4_^+^ concentrations and unaffected N-NO_3_^−^ concentrations suggest that BNF was enhanced by *A. brasilense* co-inoculation. Ureide and N-NH_4_^+^ are the main N forms transported by the xylem after symbiotic (ureide) and associative BNF (N-NH_4_^+^) [46,47]. Moreover, the isotopic technique analysis showed that the co-inoculation provided greater N accumulation in the shoot and grain specifically derived from soil and other sources (highlighting BNF), and had no effect on ^15^N fertilizer recovery. Although we did not specifically evaluate N derived from the atmosphere (NDFA), it has been reported that between 50 and 90% of total N uptake in annual legume crops is derived from BNF [8,15,16,48,49]. Under greenhouse conditions, Brito et al. [50] verified that approximately 93% of N accumulated in cowpea came from symbiotic fixation, 1.2% came from N-fertilization (N rate: 27 kg N ha^−1^) and 5.8% came from soil (a typical tropical soil). Biological N fixation is a high energy-demanding process from the host plant to provide energy to the microsymbiont and carbon (C) skeletons for the N assimilation processes [51,52]. The energy invested in N acquisition via BNF is higher than its uptake from the soil with N-NO_3_^−^ [12]. Nonetheless, according to Kaschuk et al. [53] the sink of photoassimilation caused by the symbiosis (*Bradyrhizobium* sp.) can stimulate an increase of up to 28% in the photosynthetic rate in soybean, thus compensating for any losses. As N is involved in the synthesis of chlorophylls and Rubisco, BNF also increases the photosynthetic efficiency by supplying N to the host plant [12,51,52,54].

Although evidence that N_2_ fixation contributes to the N balance of plants has been reported, many studies have shown that the contribution of N_2_ fixation by *Azospirillum* sp. to plants (an average increase of 12% in the total N of inoculated plants) is not the major role of *Azospirillum* sp. in plant growth promotion [30]. Additional plant growth-promoting mechanisms were proposed, such as the production and release of phytohormones (e.g., auxins, cytokinins and gibberellins), and nitric oxide to plants has been considered the major factor affecting root architecture [22,26,30]. Greater root development can enhance water and nutrient uptake from soil [55,56,57]. Nonetheless, a more vigorous root system has a greater rhizodeposition of organic forms of N and C, which can favor trophic interactions and biodiversity in the rhizosphere that overall benefit the plants [22,35]. Hence, the additive hypothesis best addresses *A. brasilense* operating principle, in which multiple mechanisms work in convergence or in a sequence determined by plant–soil–environment–bacteria interactions [30,39].

The benefits of co-inoculation on root development favored cowpea shoot development, likely as a result of the greater N accumulation observed. A greater N supply can increase the photosynthetic apparatus by increasing chlorophyll content, the amount and activity of carboxylation enzymes, total protein, sugar content, total N, and photosynthesis-related metabolites [58,59]. Chlorophyll is the main photosynthetic pigment in the Calvin cycle [60,61]. Increased chlorophyll content leads to an increased light energy uptake and light utilization capacity of plants, resulting in a greater photosynthetic CO_2_ assimilation capacity of leaves and photosynthetic quantum yields [58,59,62]. Therefore, the verified increase of chlorophyll A + B and leaf soluble protein concentrations indicates that increased photosynthetic CO_2_ assimilation likely occurred, leading to a greater shoot biomass and grain yield. Similar results were reported elsewhere when co-inoculation (*Bradyrhizobium* sp. + *A. brasilense* and *Rhizobium* sp. + *A. brasilense*) was found to improve soybean, common bean (*Phaseolus vulgaris* L.) and cowpea yield between 3 and 25% compared to single inoculation (*Bradyrhizobium* sp. or *Rhizobium* sp.) [22,34,35,36,63,64]. Garcia et al. [65] verified an average increase of 14.7% in soybean grain yield and 16.4% in total N accumulated in the grains with co-inoculation with *Bradyrhizobium diazoefficiens* and *A. brasilense* compared to the single inoculation with *Bradyrhizobium*. Furthermore, according to these authors, the co-inoculation performance was similar to or greater than that of the non-inoculated control receiving a high N fertilizer rate (200 kg N ha^−1^).

In addition, the results clearly indicate that N fertilizer application similarly influenced both single and *A. brasilense* co-inoculated treatments. In our study, N fertilizer inputs above 20 kg N ha^−1^ (40 and 80 kg N ha^−1^) reduced cowpea N uptake and growth mainly by the impairment of BNF verified by a decrease in shoot and root ureide and N-NH_4_^+^ concentrations and shoot and grain NDFS leading to a reduced soluble protein concentration in plant tissues. The inhibitory effect of N fertilizer application on BNF has been widely reported in most legume crops [21,66]. For example, a meta-analysis study conducted by Santachiara et al. [67] reported that N addition could decrease BNF efficiency in soybean by around 57% (70% under greenhouse conditions and 44% under field-grown conditions). The magnitude of the effect size is dependent on the N fertilizer rate [68]. At high N fertilization rates and under long-term [66] exposure to nitrates (more than 3 days), the inhibitory effect is large and possibly associated with an increase in O_2_ diffusion resistance inside the bacteroid [69]. Although urea is an amidic source of N, after initial hydrolysis of (NH₂)₂CO, the oxidation of N-NH_4_^+^ to N-NO_3_^−^ naturally occurs, which is a fundamental core process in the global biogeochemical N cycle [70]. There is also a complex hormone signaling between roots and *Bradyrhizobium* sp. to control all these specific regulations [71]. A recent meta-analysis study with 60 published articles regarding *A. brasilense* inoculation on maize crop (strains Ab-V5 and Ab-V6) concluded that yield responses tend towards greater increases at lower N rates (≤50 kg/ha, +8%) than at higher ones (>200 kg/ha, +3.8%), indicating that these strains are not incompatible with N fertilizers, although high N rates would negatively end up impacting crop yield [72]. In this sense, further studies regarding N fertilizer application associated with co-inoculation of PGPB under different agricultural systems should be performed so that a deeper understanding can be developed.

Previous studies performed under similar conditions (tropical soils) comparing (i) non-inoculated treatments with N fertilizer application (50 to 100 kg N ha^−1^) and (ii) *Bradyrhizobium* sp. inoculation without N fertilization in cowpea showed similar results in plant growth and yield [11,73,74]. For example, Martins et al. [11] concluded that cowpea inoculated with strains SEMIA 6462 showed similar grain yield compared to plants receiving 50 kg N ha^−1^. Soares et al. [73] found that fertilization of cowpea with 70 kg N ha^−1^ provided similar shoot biomass, shoot N accumulation, and grain yield to inoculation with SEMIA 6463. However, also according to these authors, N-fertilization reduced the dry mass and number of nodules. Similar results were observed by Ulzen et al. [74], who verified reduced dry mass and numbers of nodules and similar shoot biomass and grain yield with 100 kg N ha^−1^ application or single inoculation with strains SEMIA 6462. These studies suggest that N-fertilization above 50 kg N ha^−1^ in cowpea would negatively impact BNF leading to reduced nodulation. However, in the absence of inoculation, N fertilizer application would not impair N uptake and response to N application rates, at least up to 100 kg N ha^−1^. In addition, Brito et al. [75], studying N application rates (5.3; 40, 80, 120 and 160 kg N ha^−1^) associated with *Bradyrhizobium* sp. strain BR 2001 in cowpea, concluded that the BNF decreased between 55.6 to 81.5% as N rates increased. Furthermore, according to these authors, the symbiotic N_2_ fixation could replace 100% of the N fertilization in cowpea. Perhaps an absolute control (without any inoculation) associated with the tested N application rates would show a positive response to N rates in this study, since the BNF would not be compromised without single inoculation with *Bradyrhizobium* sp. and co-inoculation with *Bradyrhizobium* sp. and *A. brasilense*.

Our soil chemical analysis showed that the organic matter (OM) content was 21 g kg^−1^ (2%). Although the OM content in typical agricultural soil could vary from 1 to 6%, highly weathered tropical soils are known to have low OM content, usually 1 to 2% [76]. Thus, the verified OM in soil chemical analysis can be considered low relative to subtropical and temperate soils but normal for tropical soil conditions. Nonetheless, the results of this study clearly demonstrated that there is no need to supply N via mineral fertilizers when *A. brasilense* co-inoculation is performed in a cowpea crop. In the case of NPK basal fertilization, a common practice in agricultural systems worldwide, the supply of N via fertilizers using low N rates (<20 kg N ha^−1^) would not impair cowpea growth and development.

## 4. Materials and Methods

### 4.1. Field-Grown Site Characterization

The experiment was performed at the Experimental Station of São Paulo State University (−20°22;–51°22, and 335 m altitude) in Selvíria county (state of Matogrosso do Sul, Brazil) during the cowpea growing season of 2017/2018 (November to February). For the previous 30 years, the field in which the experiment was carried out had previously been cropped with annual cereal and legume crops. Furthermore, for the last 15 years, no tillage was performed in the area. Prior to cowpea, the area was cropped with maize (*Zea mays* L.). Climate classification was Aw according to Köppen–Geiger classification, tropical Savannah with dry winter. The historical mean annual maximum and minimum temperatures (in the last 20 years) were 32.4 °C and 20.8 °C, respectively, and mean seasonal rainfall (November to February) was 833.1 mm. Daily rainfall and temperature measurements during the experiment were registered according to a meteorological station located at the Experimental Station (32.8 °C and 21.2 °C maximum and minimum temperatures and seasonal rainfall 835.9 mm—Appendix A). Therefore, the field trial was conducted under typical climatic conditions related to the geographical region of this study. The climatic data can be assessed at “Canal Clima da UNESP Ilha Solteira”. Available online: https://clima.feis.unesp.br/ (accessed on 12 June 2022).

Soil was classified as Rhodic Hapludox [77] with 47% sand, 9% silt and 44% clay. The methodology described by van Raij et al. [78] was used to determine the chemical attributes of the 0–0.20 m soil depth, which are presented in Table 1. Nitrogen (total N) was determined by the semi-micro Kjeldahl method [79] (Table 1). Bradyrhizobia populations in soils were also estimated in the 0–0.20 m soil depth by the most probable number (MPN) technique [80] (Table 1).

### 4.2. Experimental Design and Treatments

The experimental treatments tested included two inoculations and four N fertilizer rates arranged in a 2 × 4 factorial randomized complete block design, with four replications. The experimental plot was composed of six rows 0.45 m in width and 5 m in length. The useful area was four central rows excluding 1 m at the end of each cowpea row.

The two levels of seeds inoculation were: (i) single inoculation with *Bradyrhizobium* sp. strains SEMIA 6462 and SEMIA 6463 (guarantee of 5 × 10^9^ colony-forming unity [CFU] per mL) (the standard inoculation for cowpea) and (ii) co-inoculation with *Bradyrhizobium* sp. and *A. brasilense* strains Ab-V5 and Ab-V6 (guarantee of 2 × 10^8^ CFU per mL). *Bradyrhizobium* sp. and *A. brasilense* strains were deposited at the “Diazotrophic and Plant Growth Promoting Bacteria Culture Collection of Embrapa Soja” (WFCC Collection # 1213, WDCM Collection # 1054) in the city of Londrina (state of Paraná, Brazil). The above-mentioned strains’ specific descriptions are available in previous studies [24,38,81]. The inoculations were performed by manually mixing the inoculant with cowpea seeds at the time of sowing. The inoculation rate used was 200 mL of liquid inoculant ha^−1^. Rhizosphere soil was randomly collected to analyze colonization by *Azospirillum* sp. and Bradyrhizobia populations at cowpea flowering. The Bradyrhizobia soil population was determined by MPN counts, with a serial dilution plant infection method [82,83]. The *Azospirillum* sp. population was also determined by MPN method, consisting of serial dilutions and inoculations performed on vials filled with semisolid NFb medium in the absence of N addition, followed by incubation for 48 h at 35 °C [84] (Appendix A).

Nitrogen fertilizer rates were: (i) control (0 kg ha^−1^); (ii) low (20 kg ha^−1^); (iii) medium (40 kg ha^−1^); and (iv) high (80 kg ha^−1^) N supply based on previous studies showing best N management for cowpea [9,11,85]. The N source used was a urea-^15^N isotope (45% of N) with an isotopic enrichment of 10% of excess ^15^N atom, for all rates. Nitrogen treatments were hand-applied without incorporation 10 days after cowpea emergence (DAE). The urea-^15^N isotope application was undertaken at the three central rows of cowpea, in a 1.0 m² area (0.74 m long spaced by 0.45 m: 0.74 m × 1.35 m = 1.0 m^2^). These microplots were carefully delimited with fiberglass flags and demarcation tape. To mitigate the N-NH_3_^+^ loss by volatilization, irrigation (14 mm of water) was performed in the experimental area within a few hours of N application.

### 4.3. Cowpea Crop Management

Cowpea cultivar “BRS Tumucumaque” was sown at a density of 26.7 viable seeds m^−2^ with a no-till drill. According to chemical soil characteristics and cowpea requirements [86], during sowing, basal fertilization application was performed for all treatments with 40 kg P_2_O_5_ ha^−1^ and 40 kg K_2_O ha^−1^ (triple superphosphate, 46% of P_2_O_5_; potassium chloride, 60% of K_2_O). No N application was performed in the basal fertilization. In addition, N rates were manually applied 10 DAE. Cowpea was cultivated from 10 November 2017 to 30 January 2018 (harvested 76 DAE). Weeds were controlled using pre and post-emergence herbicides, and insects were controlled using best management practices in cowpea.

### 4.4. Samplings, Measurements and Analysis

#### 4.4.1. Leaf, Root, Nodule and Soil Collection

From the upper third of cowpea plants (diagnostic leaf), 40 trifoliate leaves were collected during flowering (R1 phenological stage) for leaf analysis (N fractions, soluble proteins and photosynthetic pigments). In total, 20 trifoliate leaves were frozen using liquid N then stored at −80 °C in an ultra-freezer for ureides and soluble proteins analysis. In total, 10 trifoliate leaves were stored in 80% acetone to extract chlorophyll A + B. The remaining 10 trifoliate leaves were oven-dried with a forced-air drier at 60 °C for 90 h, sieved (2 mm) and kept for total and inorganic N (NO_3_^−^ and NH_4_^+^) determination. At the same time (R1 phenological stage), a side trench of approximately 0.50 m depth × 1.0 m length was gently dug for root and nodule collection, totalizing 10 plants collected (equivalent to 1 m × 0.45 m—cowpea row space, excluding the first and the last plants) and then quickly washed with deionized water to remove the excess of soil. Five roots with nodules were frozen using liquid N then stored at −80 °C in an ultra-freezer for ureides and soluble proteins analysis. The other five roots with nodules collected were oven-dried on a forced-air dryer at 60 °C for 90 h and root and nodule biomass were determined using a high precision scale (precision of 0.01 g). After root and nodule biomass were weighed, these samples were sieved (2 mm) and kept for inorganic N determination. Ten soil samples per plot were collected from depths of 0.00 to 0.20 m with a soil core sample type auger, 0.40 m in length and 0.10 m in diameter, when cowpea was at the R1 stage. These samples were then mixed, and two sub-samples were obtained. One sub-sample was air-dried, sieved (2 mm), and stored at ambient temperature (near 25 °C) [78] until analyzed for total N content. The other sub-sample was stored in a cold chamber below 4 °C until they were analyzed for N-NO_3_^−^ and N-NH_4_^+^.

The total N concentration in soil was determined following the methods of Malavolta et al. (1997), with sulfuric digestion and the semi-micro Kjeldahl analysis method. Nitrate and ammonium (N-NO_3_^−^ and N-NH_4_^+^) in soil were determined following the methods of Cantarella and Trivelin [87]. Briefly, 10 g of soil were extracted with 1 mol KCl L^−1^ (1:15 *w*/*v*), distilled with MgO (N-NH_4_^+^) and Devarda’s alloy and titrated with 2.5 mM H_2_SO_4_ (N-NO_3_^−^).

#### 4.4.2. Nitrogen Fractions in Plant Tissue and Soluble Protein Analysis

The same above-mentioned methodologies of Malavolta et al. [88] and Cantarella and Trivelin [87] were used to determine the total N concentration and N-NO_3_^−^ and N-NH_4_^+^ in plant tissues (sulfuric digestion and semi-micro Kjeldahl analysis method). Here, we have used 1 g of plant tissue that was extracted with 1 mol KCl L^−1^ (1:15 *w*/*v*) in sealed Erlenmeyer flasks and followed by 1 h of shaking in a table shaker (200 rpm). After shaking, the flasks were left to rest for a few minutes, then samples were filtered, distilled with calcinated MgO and Devarda’s alloy (N-NO_3_^−^) and titrated with 2.5 mM H_2_SO_4_. 

Ureides determination was performed by weighing leaves + stems and 1 g of roots + nodules, previously stored in 10 mL of MCW solution (60% methanol, 25% chloroform, and 15% water). The MCW solution was prepared following the methods described by Bielesk and Turner [89]. Sample extracts were centrifuged for 10 min at 4 °C and 10,000 rpm, and 5 mL of extract supernatant, 1.25 mL of chloroform, and 1.875 mL of water were mixed in a clean tube. The material was stored for 48 h until phase separation, then ureides were analyzed using the prepared extracts. Allantoic acid and allantoin analysis, which correspond to total ureides, were used as BNF indicators and quantified according to Vogels and Van der Drift [90]. One 250 μL hydrophilic portion of MCW extract aliquot was added to 20 μL phenylhydrazine (0.33%) and 250 μL NaOH (0.5 M). Samples were then vortexed and oven-heated (100 °C) for 8 min. Afterward, the samples were allowed to reach room temperature, at this point 250 μL HCl was added (0.65 N) and the material was vortexed and oven-heated (100 °C) for 4 min. After this second heating, the mixture was allowed to reach room temperature and 250 μL phosphate buffer (0.4 M; pH 7.0) + 250 μL phenylhydrazine (0.33%) were added. The samples were vortexed again, allowed to rest at room temperature for 5 min, and cooled in ice for 5 more minutes. Then, 1.25 mL HCl (37%) + 250 μL potassium ferrocyanide (1.65%) were added. After this last step, the mixture was vortexed and analyzed in a spectrophotometer using absorbance at 535 nm at room temperature.

The soluble protein extraction and evaluation were performed according to Bradford [91] by adding 5 mL potassium phosphate buffer (0.1 M; pH 7.5 + 3 mM dithiothreitol + 1 mM ethylenediaminetetraacetic acid) into the samples, after which the material was centrifuged for 30 min at 4 °C and 10,000 rpm. The supernatant provided by the extract was stored at −80 °C in a freezer. Samples were analyzed by collection absorbance readings at 595 nm using a spectrophotometer. Protein values were determined according to a bovine serum albumin calibration curve.

#### 4.4.3. Chlorophyll Pigment Analysis

The concentration of chlorophyll A + B was determined according to Lichtenthaler [92]. The plant material was collected fresh and stored in 80% acetone. For analysis, 200 μL of the extract was diluted into 1.8 mL 80% acetone and centrifuged. The analysis was performed in absorbance at the wavelengths 647 and 663 nm.

#### 4.4.4. Shoot Biomass and Cowpea Grain Yield

At the time of harvest (76 DAE—cowpea physiological maturity), the shoot biomass was determined by harvesting 12 plants in 0.45 m^2^ (1.0 m × 0.45 m—cowpea row space) at ground level. The plant material was oven-dried (60 °C) for 90 h in a forced-air dryer. Then, the samples were weighed and shoot biomass (kg ha^−1^) was determined. Cowpea grain yield was determined by harvesting the pods in the four central rows, each with a 5 m row length. After harvest, the grain dry weight was determined and adjusted to 13% (wet basis).

#### 4.4.5. Total N Accumulation and ^15^N-Fertilizer Recovery in Cowpea Shoot and Grain

We have considered as shoot: leaf, stem and pods without grains; and grain: only the cowpea grain. At harvest time, four cowpea plants were sampled from the centerline of the microplots. The separated subsamples (shoots and grains) were oven-dried (60 °C) for 90 h then ground (2 mm sieve) and weighed. Nitrogen concentration and ^15^N abundance (% in atoms) were determined according to Barrie and Prosser [93], using a mass spectrometer interfaced with an elemental analyzer for N (Isotope Ratio Mass Spectrometry (IRMS)—model HYDRA 20-20 ANCA–GLS, Sercon, Cheshire, UK). As is standard for ^15^N analyses, a control sample with a natural variation of the stable isotope ^15^N (0.366%) was used for every ten samples analyzed. To avoid contamination of atmospheric air gases, sample analyses were carried out under high vacuum conditions. Therefore, the isotopic abundance of the applied urea-^15^N and the natural variation of the stable isotope ^15^N (0.366%) were considered in the calculation of the recoveries. Additionally, non-enriched plants were sampled to determine the background ^15^N abundance in the fertilized and unfertilized soils, in order to account for possible small variations in the standard values of natural ^15^N abundance [94,95]. The sequence of procedures and equations was also assessed by Paulo et al. [70]. Thus, the accumulated N in shoots and grains was obtained by the product of total N concentration in tissue and produced biomass.

The N percentage in plant and/or soil derived from ^15^N labeled fertilizer (NPDFF, %) was calculated following Equation (1):(1)$$\mathrm{NPDFF}=\frac{a}{b}\times 100\;$$
where: “*a*” represents the percentage of atoms of ^15^N in excess in plant and/or soil; “*b*” represents the percentage of atoms of ^15^N in excess in ^15^N labeled fertilizer [96]. The quantity of N derived from fertilizer (shoot, SNDFF; and grain, GNDFF), in kg ha^−1^ was calculated considering NPDFF and accumulated N in cowpea shoot and grain.

The quantity of N in plants derived from soil and other sources (e.g., BNF) in cowpea shoots and grains (SNDFS and GNDFS, kg ha^−1^) was calculated following Equation (2):(2)NDFS=NA−NDFF
where NA represents the N accumulated in cowpea shoots and grains (SNDFS and GNDFS, respectively) and NDFF is the quantity of N in the shoots or grains derived from the fertilizer.

The ^15^N fertilizer recovery in shoots and grains (%) was calculated following Equation (3):(3)$$15_{N}\mathrm{fertilizer}\;\mathrm{recovery}=\frac{\mathrm{NPDFF}\;\times \;N\;\mathrm{shoot}\;\mathrm{or}\;N\;\mathrm{grain}\;}{N\;\mathrm{level}}\times 100$$
where NPDFF represents the percentage of N derived from fertilizer; N shoot and N grain represent the total quantity of N in cowpea shoots and grains, respectively; N level represents the quantity of N fertilizer applied (^15^N isotope) [96].

### 4.5. Statistical Analysis

All data were initially tested for Levene’s homoscedasticity test (*p* ≤ 0.05) and normality using a Shapiro–Wilk test, which showed the data to be normally distributed (*W* ≥ 0.90). Data were submitted for analysis of variance (F test). When a significant main effect or interaction was observed by the F test (*p* ≤ 0.05), Tukey’s test (*p* ≤ 0.05) was used for comparison of means of inoculations, N fertilizer rates and their interactions using the ExpDes package in R software [97].

## 5. Conclusions

Our findings showed that co-inoculating *Bradyrhizobium* sp. + *Azospirillum brasilense* had a significant positive influence on root development, increased root biomass, and increased N-content in the roots (mainly as ureide and N-NH_4_^+^). This significant positive effect likely was the key mechanism for the observed increase in root and leaf soluble proteins, improved cowpea growth, and increased cowpea yield.

The results of this study clearly demonstrated that there is no need for the supplementation of N via mineral fertilizers when *A. brasilense* co-inoculation is performed in a cowpea crop. However, even in the case of an NPK basal fertilization, N rates applied should remain below 20 kg N ha^−1^. Nitrogen rates applied higher greater than 20 kg N ha^−1^ will likely impair cowpea growth and development. New studies need to be designed to evaluate the effects of agronomic practices, climate change and plant growth-promoting bacteria co-inoculation under different agricultural systems. The combination of some refined scientific techniques (e.g., metabolomics, isotopic and molecular techniques, among others) is needed to deepen the understanding of the multiple mechanisms theory and the benefits of plant growth-promoting bacteria to the soil–plant–environment microbiome.

## Figures and Tables

**Figure 1 plants-11-01847-f001:**
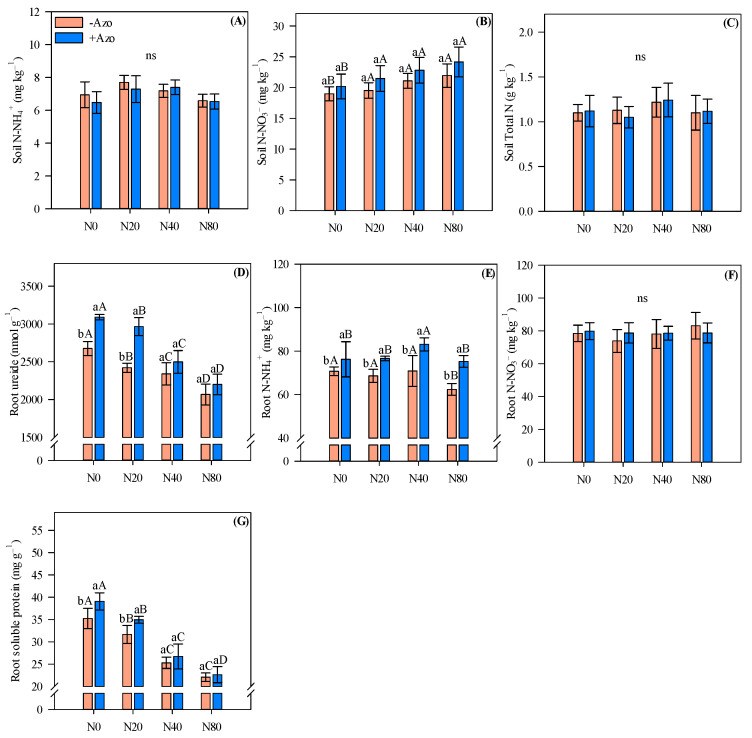
Soil ammonium (N-NH_4_^+^) (**A**), nitrate (N-NO_3_^−^) (**B**) and total N content (**C**), root ureides (**D**), root N-NH_4_^+^ (**E**), root N-NO_3_^−^ (**F**) and root soluble protein (**G**) concentrations in cowpea affected by inoculations and N fertilizer rates. Different letters indicate significant differences between treatments according to Tukey’s test (*p* ≤ 0.05). Interaction between inoculations and N levels: Lowercase letters indicate significant differences between inoculations, and uppercase letters indicate significant differences between N levels according to Tukey’s test (*p ≤* 0.05). ns = non-significant. Error bars indicate standard deviations (*n* = 4). −Azo and +Azo refer to the single inoculation with *Bradyrhizobium* sp. and co-inoculation with *Bradyrhizobium* sp. + *A. brasilense,* respectively; N0, N20, N40 and N80 refer to absence, 20, 40 and 80 kg N ha^−1^ applied.

**Figure 2 plants-11-01847-f002:**
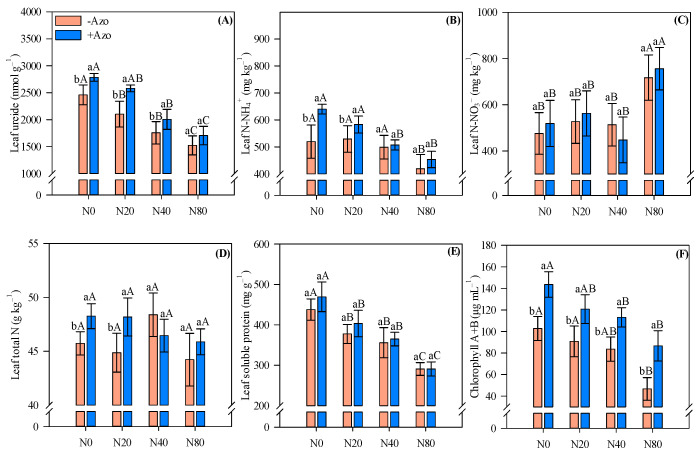
Leaf ureides (**A**), ammonium (N-NH_4_^+^) (**B**), nitrate (N-NO_3_^−^) (**C**), total N (**D**), soluble protein (**E**) and chlorophyll A + B (**F**) concentrations in cowpea affected by inoculations and N fertilizer rates. Different letters indicate significant differences between treatments according to Tukey’s test (*p* ≤ 0.05). Interaction between inoculations and N levels: Lowercase letters indicate significant differences between inoculations, and uppercase letters indicate significant differences between N levels according to Tukey’s test (*p ≤* 0.05). Error bars indicate standard deviations (*n* = 4). −Azo and +Azo refer to the single inoculation with *Bradyrhizobium* sp. and co-inoculation with *Bradyrhizobium* sp. + *A. brasilense*, respectively; N0, N20, N40 and N80 refer to absence, 20, 40 and 80 kg N ha^−1^ applied.

**Figure 3 plants-11-01847-f003:**
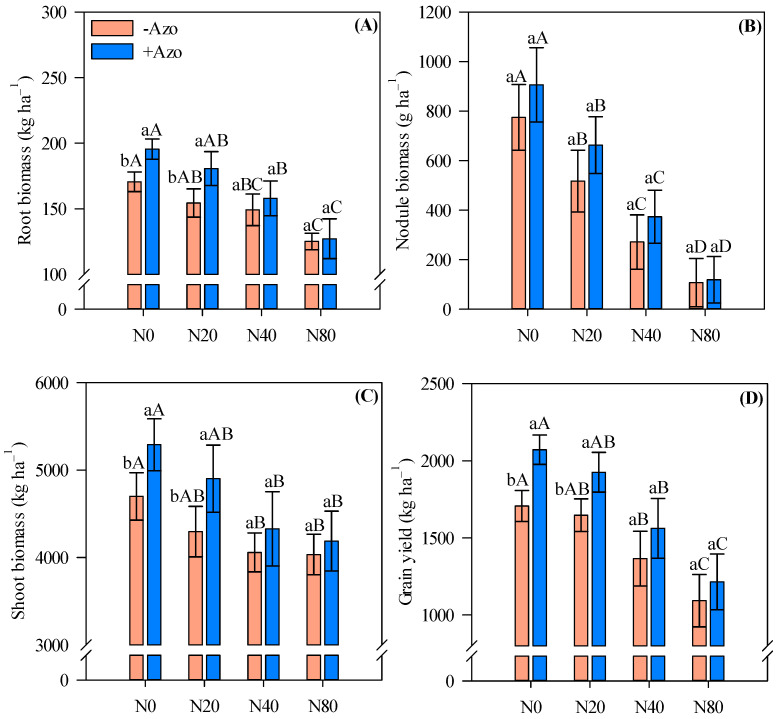
Root (**A**), nodule (**B**) and shoot (**C**) biomass and cowpea grain yield (**D**) affected by inoculations and N fertilizer rates. Different letters indicate significant differences between treatments according to Tukey’s test (*p* ≤ 0.05). Interaction between inoculations and N levels: Lowercase letters indicate significant differences between inoculations, and uppercase letters indicate significant differences between N levels according to Tukey’s test (*p ≤* 0.05). Error bars indicate standard deviations (*n* = 4). −Azo and +Azo refer to the single inoculation with *Bradyrhizobium* sp. and co-inoculation with *Bradyrhizobium* sp. + *A. brasilense,* respectively; N0, N20, N40 and N80 refer to absence, 20, 40 and 80 kg N ha^−1^ applied.

**Figure 4 plants-11-01847-f004:**
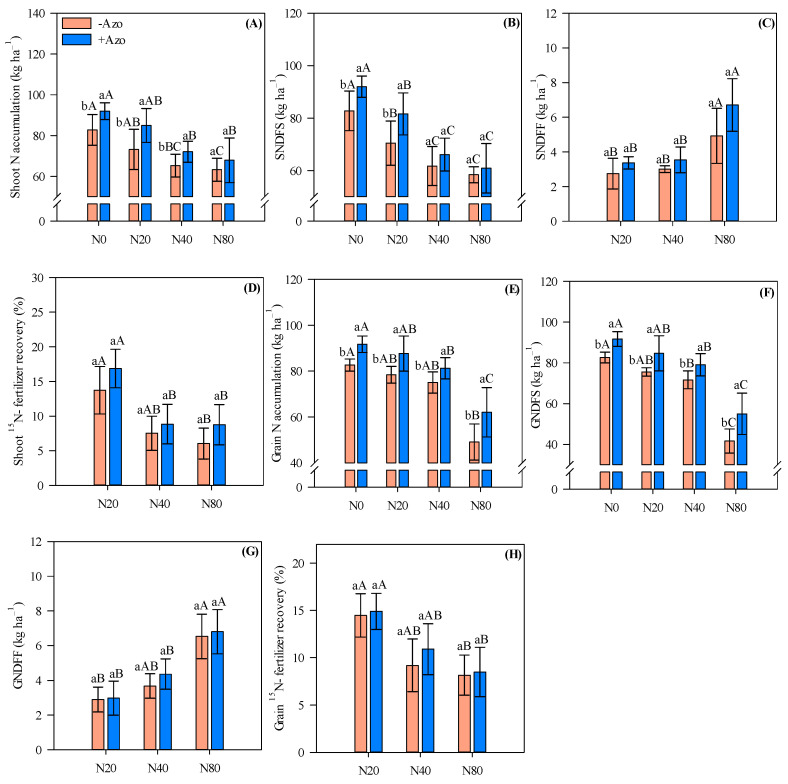
Cowpea N accumulation (**A**), N derived from soil and other sources (SNDFS) (**B**), N derived from fertilizer (SNDFF) (**C**) and ^15^N fertilizer recovery in shoots (**D**), N accumulation (**E**), N derived from soil and other sources (GNDFS) (**F**), N derived from fertilizer (GNDFF) (**G**) and ^15^N fertilizer recovery (**H**) in grain affected by inoculations and N fertilizer rates. Different letters indicate significant differences between treatments according to Tukey’s test (*p* ≤ 0.05). Interaction between inoculations and N levels: Lowercase letters indicate significant differences between inoculations, and uppercase letters indicate significant differences between N levels according to Tukey’s test (*p ≤* 0.05). Error bars indicate standard deviations (*n* = 4). −Azo and +Azo refer to the single inoculation with *Bradyrhizobium* sp. and co-inoculation with *Bradyrhizobium* sp. + *A. brasilense,* respectively; N0, N20, N40 and N80 refer to absence, 20, 40 and 80 kg N ha^−1^ applied.

**Table 1 plants-11-01847-t001:** Soil chemical attributes and Bradyrhizobia populations in soil in 0–0.20 m layer before field trial beginning.

Soil Chemical Attributes	0–0.20 m Layer
Total N	1.04 g kg^−1^
P (resin)	39.0 mg kg^−1^
S (SO_4_)	30.0 mg kg^−1^
Organic matter	21.0 g kg^−1^
pH (CaCl_2_)	5.1
K	2.3 mmol_c_ kg^−1^
Ca	31.0 mmol_c_ kg^−1^
Mg	33.0 mmol_c_ kg^−1^
H + Al	34.0 mmol_c_ kg^−1^
Al	0.0 mmol_c_ kg^−1^
B (hot water)	0.23 mg kg^−1^
Cu (DTPA)	3.7 mg kg^−1^
Fe (DTPA)	25.0 mg kg^−1^
Mn (DTPA)	30.1 mg kg^−1^
Zn (DTPA)	1.7 mg kg^−1^
Cation exchange capacity (pH 7.0)	100.3 mmol_c_ kg^−1^
Base saturation (%)	66
Bradyrhizobia populations in soil	3.5 × 10^4^ cells g^−1^ soil

Number of repetitions (n) = 20, DTPA = diethylenetriaminepentaacetic acid.

## Data Availability

All data generated or analyzed during this study are included in this published article.

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
