# Peer review of "Co-Inoculation with Azospirillum brasilense and Bradyrhizobium sp. Enhances Nitrogen Uptake and Yield in Field-Grown Cowpea and Did Not Change N-Fertilizer Recovery"

_plants, 2022, doi:10.3390/plants11141847_

Round 1
Reviewer 1 Report
The title is clear, concise and explanatory
The abstract is well structured and clear,
The introduction is correct with well-defined objectives.
The material and methods are appropriate to address the stated objectives.
The conclusions are based on the results shown.
References: suitable
Author Response
Dear editor and reviewer 1,
The point-by-point response letter is attached in the system.
Our most sincere gratitude to you and the reviewers who took time from their busy schedule to help us making this manuscript a better paper. We hope that we have answered every inquiry to your satisfaction and also hope that you will find this version of publishable quality. We hope that this version has met the expectations of the reviewer.

Reviewer 2 Report
The aim of this study is to explore the effects of Azospirillum brasilense and Bradyrhi- zobium sp. co-inoculation coupled with N application on soil N levels and N in plant, photosynthetic pigments, cowpea plant biomass and grain yield. Isotopic technique was employed to evaluate 15N-fertilizer recovery and derivation. The field experiment results showed that co-inoculation with Bradyrhizo- bium sp. + A. brasilense significantly increased cowpea N uptake and grain yield compared to the standard inoculation with Bradyrhizobium sp. specifically derived from soil and other sources without affect 15N-fertilzer recovery and there is no need for the supplementation of N via mineral fertilizers when A. brasilense co-inoculation is performed in a cowpea crop.
In general, the manuscript is well written and the experiment design is good and results are significant, which also have a important application values.
Miner points:
1. How did the authors choose the three N application level?
2. The first letter of the key word should be capital;
3. For the Fig 1A, there is no significant differences for the N application neither for the inoculation, then, this is no need put any letter aA, which make readers to more difficult to understand it. For the rest of the figure , it should do the same to make the figure simple and easy to understand.
Author Response
Dear editor and reviewer 2,
The point-by-point response letter is attached in the system.
Our most sincere gratitude to you and the reviewers who took time from their busy schedule to help us making this manuscript a better paper. We hope that we have answered every inquiry to your satisfaction and also hope that you will find this version of publishable quality. We hope that this version has met the expectations of the reviewer.

Reviewer 3 Report
The manuscript by Galindo and co-workers is basically sound and the conclusions drawn are supported by the results. English usage could be improved to make reading and comprehension easier.
Questions and comments
Lines 99-101, as a justification for the work, this is sort of weak: "...to better aid in completing our research objectives." could apply to just about any research project.
Line 109. I may misunderstand here. "Similar trends..." are claimed, but Fig. 1 graphs D, E and to some extent G seem to show much more difference between single and dual inocula treatments in comparison to the other graphs, so don’t think “similar trends” is accurate.
Line 239. In the Discussion, no need to repeat the numerical values.
Line 242. "...slight increase in nodulation." This might be interpreted as meaning the formation of more nodules rather than greater nodule biomass.
Around line 252. If 93 % of N accumulated by cowpea comes from symbiotic nitrogen fixation, why is a fairly significant amount of chemical fertilization needed at all (the need for chemical fertilization is described in the Introduction).
Paragraph starting line 262. Does Azo inoculation cause changes in root architecture that might promote nodulation by rhizobia (eg, stimulate root hair generation or something)?
Lines 247 and 348. Low N fertilization "would not impair cowpea growth", but is it needed?
Despite these fairly minor points, I think the manuscript made most of its points clearly.
Author Response
Dear editor and reviewer 3,
The point-by-point response letter is attached in the system.
Our most sincere gratitude to you and the reviewers who took time from their busy schedule to help us making this manuscript a better paper. We hope that we have answered every inquiry to your satisfaction and also hope that you will find this version of publishable quality. We hope that this version has met the expectations of the reviewer.
